# Clinical and Genetic Features of Dravet Syndrome: A Prime Example of the Role of Precision Medicine in Genetic Epilepsy

**DOI:** 10.3390/ijms25010031

**Published:** 2023-12-19

**Authors:** Hueng-Chuen Fan, Ming-Tao Yang, Lung-Chang Lin, Kuo-Liang Chiang, Chuan-Mu Chen

**Affiliations:** 1Department of Pediatrics, Tungs’ Taichung Metroharbor Hospital, Wuchi, Taichung 435, Taiwan; fanhuengchuen@yahoo.com.tw; 2Department of Rehabilitation, Jen-Teh Junior College of Medicine, Nursing and Management, Miaoli 356, Taiwan; 3Department of Life Sciences, Agricultural Biotechnology Center, National Chung Hsing University, Taichung 402, Taiwan; 4Department of Pediatrics, Far Eastern Memorial Hospital, New Taipei City 220, Taiwan; mingtao.yang.tw@gmail.com; 5Department of Chemical Engineering and Materials Science, Yuan Ze University, Taoyuan 320, Taiwan; 6Department of Pediatrics, School of Medicine, College of Medicine, Kaohsiung Medical University, Kaohsiung 807, Taiwan; lclin@kmu.edu.tw; 7Department of Pediatrics, Kaohsiung Medical University Hospital, Kaohsiung Medical University, Kaohsiung 807, Taiwan; 8Department of Pediatric Neurology, Kuang-Tien General Hospital, Taichung 433, Taiwan; lambier.tw@yahoo.com.tw; 9Department of Nutrition, Hungkuang University, Taichung 433, Taiwan; 10The iEGG and Animal Biotechnology Center, and Rong Hsing Research Center for Translational Medicine, National Chung Hsing University, Taichung 402, Taiwan

**Keywords:** dravet syndrome, developmental epileptic encephalopathy, genetics, targeted therapy, epilepsy syndrome

## Abstract

Dravet syndrome (DS), also known as severe myoclonic epilepsy of infancy, is a rare and drug-resistant form of developmental and epileptic encephalopathies, which is both debilitating and challenging to manage, typically arising during the first year of life, with seizures often triggered by fever, infections, or vaccinations. It is characterized by frequent and prolonged seizures, developmental delays, and various other neurological and behavioral impairments. Most cases result from pathogenic mutations in the sodium voltage-gated channel alpha subunit 1 (*SCN1A*) gene, which encodes a critical voltage-gated sodium channel subunit involved in neuronal excitability. Precision medicine offers significant potential for improving DS diagnosis and treatment. Early genetic testing enables timely and accurate diagnosis. Advances in our understanding of DS’s underlying genetic mechanisms and neurobiology have enabled the development of targeted therapies, such as gene therapy, offering more effective and less invasive treatment options for patients with DS. Targeted and gene therapies provide hope for more effective and personalized treatments. However, research into novel approaches remains in its early stages, and their clinical application remains to be seen. This review addresses the current understanding of clinical DS features, genetic involvement in DS development, and outcomes of novel DS therapies.

## 1. Introduction 

### 1.1. Epileptic Encephalopathies (EEs) and Developmental Encephalopathies (DEs)

The International League Against Epilepsy (ILAE) defines an EE as “the epileptiform activity itself contributes to severe cognitive and behavioral impairments beyond that expected from the underlying pathology alone (such as a cortical malformation)” [1]. EEs are conditions that can worsen over time, be seen along a spectrum of severity and across all epilepsies, and occur at any age, although most first appear in infancy and childhood [1,2]. The EE triad includes seizures (refractory), epileptiform activity on electroencephalography (EEG), and adverse effects on development, cognition, and often behavior [3]. Therefore, if patients have no epileptiform activity, they do not have an EE. Importantly, if the treatments can ameliorate the epileptiform discharges of a patient with an EE, its developmental consequences can also be improved [4]. DEs are pre-existing developmental delays or intellectual disabilities due to a non-progressive brain state, and the degree of disability may become more apparent with brain maturation. Patients with DEs have a higher risk of epilepsy than the general population, suggesting that developmental delays are the direct result of the underlying cause of their epilepsy. Genetic causes are also frequent in this population [3]. Due to several connections and overlaps between EEs and DEs, the term developmental and epileptic encephalopathies (DEEs) was coined in the formal 2017 classification revision. 

### 1.2. DEEs 

DEEs encompass a group of severe neurological disorders that combine EEs and DEs, leading to a complex clinical presentation involving seizures, developmental delays, and cognitive impairments [4]. These conditions are often associated with a high disease burden, including significant morbidity, mortality, and reduced quality of life [5].

Various factors contribute to developing DEEs, including genetic mutations, structural brain abnormalities, and metabolic disorders. In neonatal onset DEEs, nearly 60% of cases are attributable to specific causes, compared to approximately 50% of later onset DEEs [5]. Genetic mutations cause a significant proportion of DEEs, with many cases involving genes associated with synaptic function, ion channels, and neurotransmitter signaling. Advances in genetic testing, such as whole exome sequencing and targeted gene panels, have improved our understanding of the genetic landscape underlying DEEs [6]. DEEs are characterized by several key features, including variable age of onset, etiologies, seizure types, and EEG patterns, which may change as age advances; drug-resistant seizures; developmental delays and intellectual disability (ID); co-occurring motor, speech, and behavioral impairments; and a high risk of complications. In spite of the cessation of seizures, their outcomes may remain very poor, and early death has been reported in some cases. In addition, various epilepsy syndromes fall within the DEE category (Table 1). However, a key component of the concept is that the amelioration of the epileptiform activity may potentially improve the developmental consequences of the disorder [3,4,7].

## 2. Dravet Syndrome (DS)

DS, also known as severe myoclonic epilepsy of infancy, epilepsy with polymorphic seizures, and polymorphic epilepsy of infancy, is a severe drug-resistant form of DEE that is both debilitating and challenging to manage [8]. It is a rare, genetic epileptic disorder typically arising during the first year of life, with seizures often triggered by fever, infections, or vaccinations [9]. Over 80% of cases result from pathogenic mutations in the sodium voltage-gated channel alpha subunit 1 (*SCN1A*) gene, which encodes a critical voltage-gated sodium channel subunit involved in neuronal excitability [10]. While DS’s exact pathophysiological mechanisms remain incompletely understood, its underlying genetic basis contributes to the refractory nature of seizures and the diversity of associated developmental and cognitive impairments [11]. Patients may experience frequent and prolonged seizures, developmental delays, intellectual disabilities, motor impairments, and behavioral challenges [12]. DS substantially burdens affected individuals and their families with a broad spectrum of clinical features and multi-systemic comorbidities [13]. Patients with DS have a significantly increased risk of premature mortality, with sudden unexpected death in epilepsy (SUDEP) being the leading cause. 

With the promising results of gene therapy for *SCN1A*-mutated experimental models, prompt identification and accurate definition of DS have become crucial to optimize patients’ outcomes. However, DS typically presents with a broad range of phenotypes, while molecular identification may miss 20% of the patients with DS but without an *SCN1A* mutation. A recent report found that traditional DS phenotypic spectrums and diagnostic criteria may limit the accurate diagnosis of patients [14]. For example, it was found that the age of onset extended to 19 months of life instead of the previously established 12 months. In addition, fever-related seizures were less prevalent than previously thought. In contrast, tonic–clonic seizures were the most common seizure form; hemiclonic seizures were previously believed to be the most common form in DS [14].

Precision medicine, a medical approach that tailors treatment to patients’ individual characteristics, has the potential to significantly improve DS management. Several gene therapy approaches have been proposed and investigated in recent years, including antisense oligonucleotides (ASOs) and adeno-associated virus (AAV)-mediated gene replacement [15,16]. In addition, pharmacological approaches addressing the imbalance between excitatory and inhibitory neurotransmissions caused by SCN1A dysfunction have been investigated, such as stiripentol, fenfluramine, and cannabidiol (CBD) [8]. As more treatment options tailored to DS become available, we aim to comprehensively review the current understanding of clinical DS features, genetic involvement in DS development, and outcomes of novel DS therapies.

### 2.1. Epidemiology

Data indicate that DS has a global incidence of approximately 6.5 per 100,000 live births [9,17,18,19]. However, it may vary substantially across ethnicities and geographical regions. For example, in the United States (US), a population-based study found that the incidence of DS was one per 15,700 live births, with six of the eight identified cases having a de novo *SCN1A* missense mutation. This finding indicates that DS due to an *SCN1A* mutation is twice as common in the US as previously thought [19]. Data from Denmark also indicate an incidence of DS of one per 22,000 live births, with fifteen of the seventeen patients having an *SCN1A* mutation [20]. In Germany, the 10-year prevalence of DS was estimated to be 4.7 per 100,000 people [21]. The estimated prevalence is 1/20,000–1/30,000 in France [22] and 1/45, 700 children aged under 18 years in Sweden [23]. DS with a confirmed *SCN1A* mutation has an incidence of at least 1/40 per 900 births in the UK [24], and 1/15 per 500 live births in Scotland [18]. While rare, DS accounts for approximately 2–5% of all childhood epilepsies [25]. Males and females are equally affected [26]. Nevertheless, more extensive population-based studies are needed to accurately determine the prevalence and distribution of DS across different populations.

### 2.2. Clinical Presentation of DS

#### 2.2.1. Neurodevelopment in DS Starts before the Age of Two Years and Continues to Worsen

Most children with DS show neurodevelopmental delay, often having delays in gross motor, fine motor, language, and psychomotor function, leading to global neurodevelopmental delays that majorly impact affected children [27]. Generally, affected children are almost normal at baseline. Motor delays slowly appear in them before the age of six years and rapidly worsen with age [24,28,29,30,31,32]. While gross motor milestone achievement is diverse in DS, a report assessing children with DS aged over two years showed that approximately half had significant delays in independent sitting and walking [31]. These results are consistent with our experiences and several studies, despite differences in different domains, irrelevant to the different courses of their seizure histories [30,32]. Crouch gait, an abnormal walking pattern, is mainly caused by spasticity characterized by excessive knee and hip flexion and ankle dorsiflexion during the stance phase [33]. One study that performed formal gait analysis on patients with DS aged 2–34 years found crouch gait in 88% of adolescents aged > 13 years, 50% of children aged ≥ 6 years, and no children aged < 6 years [34]. Affected children are reported to frequently show neurological impairments, such as ataxia, extrapyramidal signs, myoclonus [31], and tremors [11]. Mounting evidence shows that fine motor skills are even more affected than gross motor skills in affected children [28,30]. Verheyen et al. reported that 100% of children with DS developed gross motor delay, while 77% developed fine motor delay [30,32]. While language may be less affected in some patients, their social abilities are usually maintained [35]. 

A significant delay in psychomotor development is one characteristic of DS that is primarily moderate to severe [32,36,37,38]. Poorer cognitive outcomes are reportedly associated with higher frequencies of seizures [32], status epilepticus (SE), and interictal EEG abnormalities in one-year-olds [24], and the early appearance of myoclonus, the absence of seizures [39], and motor dysfunction, including ataxia, dyskinesia, hypotonia, and spasticity [24]. Pathogenic *SCN1A* mutations have also been reported to be associated with poor cognitive outcomes [40]. Many children with DS experience a decline in their intelligence quotient (IQ) [41]. One study showed that ID was typically apparent in children with DS by age 18–60 months [42]. Another study involving 21 children with DS showed a decline in IQ over time, with no child having a normal IQ beyond the age of six years. These children also showed signs suggestive of attention deficit hyperactivity disorder (ADHD) [43]. Nabbout et al. reported that IQ was generally normal before age two but declined after three years, and ADHD was also prevalent [40]. A quarter of children with DS show autistic behavior with poor eye contact and stereotypes [44]. The global decline in DS may be due to a combination of factors, including seizure effects, the underlying genetic defect, and medication effects. Seizures can damage the brain, and the SCN1A gene mutation can lead to abnormal brain development. While seizures may lessen in the third phase, behavioral, communication, and developmental manifestations may persist from infancy to adulthood [31]. Medications for treating seizures, (such as sodium channel blockers: (e.g., carbamazepine, eslicarbazepine, lamotrigine, oxcarbazepine, phenytoin, and vigabatrin) can also have cognitive side effects [45]. Whether early and proper control of seizures in children with DS can stop their development of psychological disorders, such as ADHD and autistic-like behavior, deserves further study. 

#### 2.2.2. Triggering Factors

DS is a severe form of DEE that typically begins during the first year of life. In children, common triggers include fever, infection, sun exposure, flashing lights, high ambient temperature, visual patterns, hot-water baths, intense physical exercise, eating, and over excitement [11,46,47]. Compared to children with DS, adults with DS may be much less susceptible to seizures induced by hyperthermia and other triggers that induced seizures in childhood. However, seizures tend to exacerbate in older children and adults with the use of sodium channel blockers [42].

#### 2.2.3. Seizure Patterns

Initially, otherwise healthy children may experience frequent, prolonged hemiclonic or generalized seizures [11]. Patients can suffer from recurring febrile and afebrile seizures that impact different body parts as they grow. Various seizure forms can emerge between the ages of one and four years, including myoclonic, atypical absence, focal, and generalized tonic–clonic seizures. Focal seizures might progress to focal motor or bilateral convulsive seizures. However, tonic seizures are uncommon in individuals with DS [12,48].

In a Chinese retrospective study, 77% of children with DS and SCN1A mutations had seizure onset before seven months, with 72% experiencing febrile seizures lasting >15 min and 67% having multiple febrile seizures within a day [49]. Li et al. found that the age of onset extended to 19 months of life and that tonic–clonic seizure was the most common seizure form [14]. Another report found that seizure onset under seven months; frequent, prolonged seizures; partial, focal, and myoclonic seizures; and hot water-induced seizures significantly correlated with DS [50]. 

Convulsive and nonconvulsive SE frequently occur in DS. Nonconvulsive status epilepticus (NCSE), also called “obtundation status epilepticus”, is reported to be in about 40% of patients with DS in several series and can be life-threatening [41,51]. Moreover, NCSE can present subtle symptoms that make it difficult to recognize [11,52]. As epilepsy progresses, children with DS often experience a decrease in seizure frequency and intensity as they transition into adolescence and adulthood. While fever sensitivity remains, its effects lessen over time. Myoclonic, atypical absence, focal seizures with altered awareness, and SE become less frequent in adults [53]. In adulthood, the most common seizure type is generalized tonic–clonic, which may have a focal origin and primarily occur during sleep. Tonic vibratory or clonic movements may occur in asymmetrical or bilateral patterns [11]. In a study of five patients with DS who transitioned into adulthood, their EEG recording showed that three had focal seizures, with or without progression into bilateral convulsive seizures. However, obtundation and tonic events occurred in the other two patients [53]. In a report that followed 53 children with DS for up to 14 years, seizure frequency was weekly in more than two-thirds of cases, while monthly and daily frequencies were observed in 13% each [54]. Several studies have shown that the frequency and intensity of seizures may gradually decrease during adolescence and into adulthood but vary among individuals [38,55,56,57,58,59]. In addition to age-related maturation effects, whether the roles of sex-related hormones contribute to the evolution of seizures in DS needs further investigation.

#### 2.2.4. EEG Evolution in DS

During the first year of life, EEGs are typically normal for patients with DS. As the disease evolves, the EEG may remain normal or become slower while sleep patterns generally remain intact. Common observations include generalized spike-and-wave patterns, polyspike waves, and multifocal spikes. However, few studies have comprehensively examined the development of EEG findings in DS over time [11]. In an EEG study of 22 children with DS during the first five years after diagnosis, all children had a normal EEG background at onset, but 27% showed background slowing after six months. At the onset of seizures, epileptiform discharges were present in 27% of the children, increasing to 64% by the five-year follow-up. In addition, while almost 57% of the children had multifocal epileptiform after five years, approximately 30% and 14% had focal epileptiform discharges and generalized epileptiform, respectively. The rate of photoparoxysmal response increased to 41% from a baseline of 9% after five years [60].

Another report showed that 81% of children with DS had abnormal EEG results; 25% had an abnormal EEG at the time of seizure onset. Generalized spike-wave discharges were the most common epileptiform pattern [61]. Other reports confirmed these findings and showed that epileptiform discharges were present in most patients, with multifocal epileptiform discharges being the most common [62]. Notably, Genton et al. described varying EEG backgrounds and epileptiform activity patterns among 24 patients with DS, with photosensitivity and pattern sensitivity typically disappearing by 20 years of age [53]. The diameter of the human optic nerve varies across developmental stages. It is small at birth, enlarges during childhood, and remains constant in adulthood [63]. The growth of the optic nerve is generally correlated with the evolution of photosensitivity and pattern sensitivity in DS, suggesting that gene therapy for DS should be given before adulthood. 

#### 2.2.5. Neuroimaging in DS

Structural brain imaging using magnetic resonance imaging (MRI) is mostly normal in DS; however, abnormalities have been reported in some cases. A study reviewing 120 Italian children with DS found cortical development malformations in four [64]. In another study of eighteen children with DS who underwent MRI after age three, seven showed hippocampal sclerosis or a loss of gray-white distinction in the temporal lobe [65]. A comparative study of nine patients (three adults and six children) with DS and SCN1A mutations and nine seizure-free controls showed that patients with DS had globally reduced gray and white matter volumes. However, total intracranial volume has been reported to decrease significantly with age [66]. Another study showed that some patients with DS had cerebral or cerebellar atrophy or hippocampal sclerosis [67].

#### 2.2.6. DS Comorbidities 

DS is often associated with various comorbidities, such as blood, cardiac, digestive, endocrine, infection, neurodegenerative, orthopedic, and sleep impairments (Figure 1) [13]. 

##### Psychomotor

The reported incidence of ID (67–100%), autism (0–100%) [68,69,70,71,72,73], and ADHD (0–66%) [37,74] in patients with DS are substantially higher than the general population [75]. 

##### Blood

Approximately one-third of patients with DS were reported to have low platelets, vitamin D deficiency, and iron deficiency [13].

##### Cardiac 

Since the brain and heart essentially share a similar sodium channel condition, neural sodium channel mutations may directly or indirectly affect cardiac functions, leading to conditions such as variable heart rate variability [76], arrhythmia [77,78], and changes in the heart structure [13].

##### Digestive

Gastrointestinal problems are commonly reported symptoms in DS [13]. A cross-section study found that more than one-third of patients with DS had dysphagia-related symptoms, such as impaired chewing, impaired swallowing, choking, and drooling; more than 40% of patients with DS had behavioral symptoms, such as picky eating and distracted during meals, and more than half of patients with DS had gastrointestinal symptoms, such as constipation and loss of appetite [79]. Stiripentol, a very effective AED against seizures in DS, was reported to be linked to worsening gastrointestinal symptoms in patients with DS [80].

##### Endocrine

Endocrine abnormalities associated with DS are very rare. Only one retrospective study involving 68 children with DS showed decreases in mean height and weight that were insensitive to age, sex, family history, and antiepileptic drugs (AEDs). Furthermore, they also found that some had low thyroid-stimulating hormone (TSH), insulin-like growth factor 1(IGF-1), and testosterone levels, supporting the idea that endocrine dysfunction is related to DS [81]. Delayed and precocious puberty was also reported in patients with DS due to SCN1A mutation [13]. 

##### Infection

Chronic infectious diseases such as bronchitis, pneumonia, and otitis media have been frequently reported in patients with DS [13]. 

##### Neurodegenerative

Dysautonomia (autonomic dysfunction) of the heart may induce critical arrhythmia, which is a risk factor associated with sudden death in several heart diseases, with numerous experimental models supporting this association [82]. In addition, dysautonomia-induced overheating may potentially trigger seizures in patients with DS [1]. However, the underlying cause of dysautonomia is unclear. In another study of 12 adults with DS, Fasano et al. identified Parkinsonism features in 91%. Two patients experienced significant improvement in their Parkinsonian symptoms after receiving levodopa treatment [83].

##### Orthopedic/Movement

Individuals with drug-resistant epilepsy (DRE) often experience osteopenia, leading to elevated fracture risk [84]. However, osteopenia risk has not been studied in patients with DS. The reported incidence of ataxia (62%) [85], crouch gait (40%) [85], and hemiparesis (4.8%) [85] are common in patients with DS. 

##### Sleep

Sleep disturbances are common in patients with DS. One study detected sleep disturbances in 75% of patients with DS [86], with another using an animal DS model supporting this finding [87]. Seizures may be one main cause of sleep disruption in patients with DS [88].

##### Eyes

Photosensitivity is common in patients with DS [13]. It may be noticed when a flash of light induces eyelid myoclonic jerks, myoclonia, absences, or generalized tonic–clonic seizures by eye closure, watching television, or a fixation on patterns [52]. One study involving 12 patients with DS found that their hand and eye coordination were significantly impaired [35]. Importantly, impaired visual functions were found to precede cognitive decline in patients with DS, and visual functions were found to continually deteriorate in patients with DS, specifically those involving a specific visuo-motor dorsal pathway [89]. In addition to visual impairment, eyelid closure may also be involved in DS. A case series reported that patients with DS presented repetitive episodes of eyelid closure, which was proposed to be an early motor trait of DS [90]. 

#### 2.2.7. SUDEP and Mortality

The incidence of SUDEP is significantly higher in patients with DS than the general epilepsy population, with an estimated incidence of 9.32 per 1000 person years [91]. The risk of mortality is notably high in patients with DS. While mortality risk is higher for patients with DS than the general population in all age groups, with a reported mortality rate of 3–21%, the highest mortality rate is observed during childhood. Previous studies have shown that the most common causes of death in patients with DS are SE, SUDEP, and drowning/injury during a seizure event [11]. Current evidence indicates that SUDEP risk is 15 times higher in DS than in other epilepsies [26]. The exact etiology of SUDEP in DS remains unclear. However, it has been hypothesized that the hyper-stimulation of parasympathetic activity after generalized seizure increases the risks of severe bradycardia and electrical heart failure. A previous animal study showed that *SCN1A*-mutant mice had brain-induced bradycardia after a seizure [92]. Other studies have proposed that *SCN1A* haploinsufficiency contributes to ventricular dysfunction and arrhythmia in mice with an *SCN1A* mutation [93,94]. Because prolonged seizures without appropriate intervention may lead to SE or increase the risk of SUDEP, the detection of seizures is considerably important. While the core features of DS remain consistent across both children and adults, adults often represent a diagnostic challenge for DS [42] since they usually have different clinical features (Table 2).

## 3. DS Diagnosis and Long-Term Prognosis 

### 3.1. Diagnostic Criteria

A large Dutch cohort study revealed that the time for making a corrective diagnosis of DS is frequently delayed, even as late as three years of age [95]. With the disease progression and exposure to inappropriate AEDs, patients with DS may experience seizure exacerbation and adverse neurodevelopment. Therefore, the diagnostic criteria for DS are critical and essential. While the diagnostic criteria for DS have evolved over time, DS remains a clinical diagnosis and should be suspected in previously healthy infants presenting with drug-resistant seizures before the first year of life, typically accompanied by fever and associated with neurodevelopmental regression. The ILAE diagnostic criteria for DS are summarized as follows [9]:Seizure onset in a previously healthy infant within the first year of life [96].A history of febrile and afebrile seizures [97].Seizures resistant to conventional AEDs [98].Prolonged, recurrent febrile, and afebrile seizures [97].Appearance of multiple seizure types, including febrile or afebrile generalized/unilateral clonic/tonic seizures, myoclonic, and atypical absence. Focal seizures progress to focal motor or bilateral convulsive seizures [99].Children show cognitive and behavioral impairments from the second year of life [97].At the onset of the first seizures, neither parents nor physicians notice any developmental delay, which insidiously appears during the second year of life [9].A family history of epilepsy. *SCN1A* gene mutations were initially identified in cases of genetic epilepsy with febrile seizures plus (GEFS+). A family history of epilepsy or febrile seizures has been noted in 15–35% of DS cases, with affected family members often showing GEFS+. DS has been documented in identical twins, infrequently in non-identical twins, and occasionally among multiple siblings within a single family. A family history of febrile seizures and epilepsy in individuals with DS and de novo SCN1A mutations implies a polygenic mode of inheritance. Additional modifier genes, such as sodium voltage-gated channel alpha subunit 9 (SCN9A), may contribute to DS manifestations [100,101]. Abnormal EEG findings are characterized by generalized or multifocal epileptiform discharges [9].Seizures with photosensitivity are more challenging to control in DS [9].Psychomotor slowing followed by crouch gait, pyramidal signs, or interictal myoclonus [9].The recent consensus strongly recommended genetic testing for patients with suspected DS [9].

### 3.2. Genetic Testing in DS

Genetic testing for DS is vital for several reasons, including early and accurate diagnosis, tailored treatment strategies, and genetic counseling for families [100]. Early DS diagnosis is crucial since it has been shown that early recognition and intervention can significantly improve patient outcomes and quality of life [102]. In addition to guiding treatment, genetic testing for DS is essential for providing accurate genetic counseling to families. Since DS is an autosomal dominant disorder with a 50% chance of inheritance for each offspring of an affected parent [48], identifying its genetic basis allows families to make informed decisions about family planning and prenatal testing. Moreover, identifying the specific *SCN1A* mutation can help predict the severity of the disease and provide prognostic information since some *SCN1A* mutations have been associated with more severe phenotypes [10].

## 4. The Role of *SCN1A* in DS

### 4.1. SCN1A Mutations in DS

The voltage-gated sodium channel (VGSC) family comprises nine isoforms (Nav1.1–Nav1.9). Structurally, these VGSCs contain one of nine large α subunits, which in humans are encoded by nine distinct genes (SCN1A-5A, and SCN7A-10A) and one of four ß subunits (Navβ1–Navβ4), which in humans are encoded by four distinct genes genes (SCN1B-4B) [103]. The α subunits have all the machinery functions for channel cell surface expression, ion conduction, voltage sensing and gating, and inactivation. The β subunits are accessories to α subunits and help regulate gating, kinetics, and channel surface density [104]. 

SCN1A mutations account for nearly 80% of DS cases [49], and more than 2584 mutations have been identified to date (Human Gene Mutation Database, http://www.hgmd.cf.ac.uk/ac/index.php (accessed on 22 November 2023)). The SCN1A gene is located on chromosome 2q24.3 and contains 27 exons. It encodes the sodium voltage-gated channel alpha subunit 1 (Nav1.1) with 2009 amino acids (~229 kDa) [105]. The Nav1.1 channel comprises four repeated domains (DI-DIV), each containing six α-helical transmembrane segments (S1–S6) spanning the cell membrane; S4 is a voltage sensor, and S5 and S6 form the pores that allow ions to pass through (Figure 2A,B) [106]. 

Similar to other VGSCs, the Nav1.1 channel maintains the balance between neuronal excitation and inhibition, shaping complex neural networks that underlie cognition, perception, and behavior [107]. The NaV1.1 channel is predominantly expressed in the central nervous system, particularly in inhibitory GABAergic interneurons, where it contributes to regulating neuronal excitability and fine-tuning synaptic transmission [106]. Dysfunction of the NaV1.1 channel has been implicated in various neurological disorders, including epilepsy, migraines, and autism spectrum disorders [108].

Mutations in the *SCN1A* gene were initially identified in cases of GEFS+. While most mutations develop due to de novo truncating or missense changes, a family history of epilepsy or febrile seizures has been noted in 15–35% of DS cases, with affected family members often exhibiting GEFS+. DS has been documented in identical twins, infrequently in non-identical twins, and occasionally among multiple siblings within a single family. The presence of unaffected or mildly affected parents can be explained by the possibility of mosaic mutations, whether somatic or germline; previous reports suggested that mosaicism is present in 7% of DS families [109,110]. A family history of febrile seizures and epilepsy in individuals with DS and de novo *SCN1A* mutations implies a polygenic mode of inheritance. Additional modifier genes, such as sodium voltage-gated channel α subunit 9 (SCN9A), may contribute to the clinical presentations of DS [100,101]. Nav1.1 channel activation depends on activating the fourth transmembrane segment (S4) of the four repeated domains (DI-DIV). On the other hand, the loop between DIII and DIV interacts with loops between S5 and S6 to inactivate the channel. SCN1A mutations affect the function of NaV1.1 differently based on the involved amino acids [111]. In return, there are obvious genotype–phenotype correlations in DS syndrome, and the disease severity can be closely correlated with the extent of mutations. Although GEFS+ is associated with less severe seizures, patients with GEFS+ and SCN1A mutations (nearly 10–15% of the patients) are characterized by hyperthermia-induced severe seizures [112]. In a previous retrospective study, patients with DS exhibited higher frequencies of truncating mutations and greater alterations in amino acid polarity compared to patients with GEFS+, supporting the involvement of SCN1A mutations in the clinical severity of DS [113]. However, it should be noted that the relationship between SCN1A mutations and phenotypes has not been fully elucidated yet. Other factors, like genetic background, may affect DS’s clinical severity. As the expression of voltage-gated sodium channels changes during the developmental process, it is expected that SCN1A expression may play a role in the developmental onset and susceptibility to febrile seizures [114,115].

**Figure 2 ijms-25-00031-f002:**
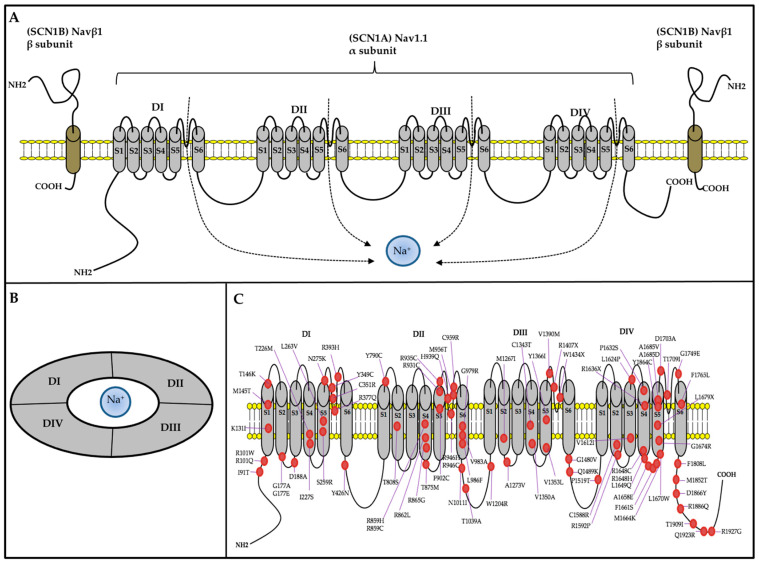
The typical structure of a voltage-gated Nav channel. (**A**) The VGSC contains α and β subunits, which are transmembrane proteins. The α subunit contains four homologous domains (DI-IV). Each domain comprises six transmembrane regions (S1–S6). S1–S4 forms the voltage-sensing domain, and S5 and S6 comprise the pore-forming domain. The β subunits have a single transmembrane segment, a short intracellular domain with a single loop. (**B**) Top view from the extracellular face and side view of the Nav channel. The four domains of the α subunit form a Na^+^ permeable pore lined by the re-entrant S5–S6 pore-loop segments. (**C**) The location of some *SCN1A* variants, which are summarized from three studies [47,116,117].

### 4.2. SCN1A-Mediated Hyperexcitability in DS: The Sodium Channel Interneuronopathy

Functional studies have shown that SCN1A mutants function differently, including loss-of-function (LOF) and gain-of-function (GOF). Among the VGSC subtypes, Na_v_1.1 is the most important VGSC subtype for initiating action potentials in bipolar GABAergic inhibitory interneurons, which have crucial inhibitory activity [118]. Consequently, SCN1A LOF variants may cause dysfunction in those inhibitory interneurons and alter neuronal excitability by reducing inhibition, leading to a predisposition to seizures [119]. AEDs, such as valproic acid (VPA), which can elevate gamma-aminobutyric acid (GABA) levels by inhibiting transaminase, may be effective. However, sodium channel blockers may aggravate seizure attacks [120]. GOF reflects a small and non-inactivating current and becomes a large and activating one, leading to neuron hyperexcitability [121]. SCN1A GOF-mutant-associated epilepsy may be controlled by the appropriate use of adequate AEDs that can inhibit sodium channels, such as phenytoin and carbamazepine. Modifying the gating properties of the SCN1A mutant can lead to an imbalance in excitatory to inhibitory neurotransmission in neural circuits, causing both SCN1A-associated epilepsies and comorbidities [102] (Figure 2C).

In addition, SCN1A haploinsufficiency decreases the availability of functional NaV1.1 channels on the cell membrane, impairing their ability to release the inhibitory neurotransmitter GABA effectively [102]. The reduced inhibitory input onto excitatory neurons disrupts the balance between excitation and inhibition in the neuronal network, resulting in hyperexcitability [122]. This heightened excitability renders the brain more susceptible to seizures, which are often triggered by external factors such as fever, stress, or sensory stimulation [11]. Several experimental models confirmed the involvement of SCN1A LOF in hyperexcitability and seizures [119,123]. In an SCN1A knock-out model, there was increased susceptibility to hyperthermia-induced seizures, which was found to arise from the reduced sodium currents in the inhibitory GABAergic interneurons. It was also noted that there is impaired inhibitory input to postsynaptic cells [122].

The hypothesis of dysregulated inhibitory interneurons in DS is further supported by the NaV1.1 expression distribution pattern. Three major classes of interneurons, distinguished by their molecular markers and functional properties, have been extensively studied: parvalbumin (PV), somatostatin (SST), and serotonin receptor 3a (5-HT3aR)-expressing interneurons [124]. PV interneurons are fast-spiking cells that predominantly target the perisomatic region of excitatory neurons [125]. They play a crucial role in regulating the temporal aspects of neural activity, such as generating gamma oscillations, which are essential for cognitive processes, like attention and memory [126]. Previous reports found that NaV1.1 expression is more predominant in the PV+ interneurons of the neocortex, hippocampus, and cerebellum, supporting the involvement of PV+ GABAergic neurons in SCN1A-related epilepsy [111]. Additionally, recent evidence highlighted that in DS models, the deletion of Nav1.1 in PV+ interneurons led to autistic-like social interaction deficits and pro-epileptic effects [127].

### 4.3. Interneuronopathy and Cognition, Developmental Impairment, and Comorbidities

#### Role of Interneuronopathy in Cognition and Development

When evaluating the impact of SCN1A mutations on cognition in DS, it is crucial to understand Nav1.1’s function in a healthy and developing brain. Research from animal models indicates that SCN1A mutations in DS impair the function of GABAergic interneurons, particularly PV+ interneurons. These cells are essential for the spatiotemporal regulation of neural network activity during cognitive processes. Thus, even a slight impairment in their function may lead to significant cognitive and behavioral disturbances, contributing to severe cognitive impairments in individuals with DS [102].

Interneurons are more diverse and complex than initially thought, serving roles beyond simply regulating excitability. They can be classified based on their target cells’ dendritic or perisomatic domain, with the latter being characterized by the expression of PV. Perisomatic-targeting PV+ interneurons are essential for regulating neural network synchrony and temporal patterns, as demonstrated by their role in gamma and theta oscillations [125,128]. These oscillations are vital for cognitive processes and require the proper function of PV+ fast-spiking interneurons, which are believed to be primarily affected by DS [111]. Gamma oscillations are a type of brain rhythm associated with cognitive functions such as sensory processing, memory, and attention. PV+ interneurons play a significant role in generating cortical gamma oscillations. In animal models of DS, disrupted gamma oscillations have been observed, which may contribute to the cognitive impairments observed in affected individuals [129].

Theta oscillations, which play a significant role in cognitive function and provide a temporal framework for neural network activity, also rely on PV+ interneurons. In the hippocampus, PV+ interneurons help generate theta oscillations, and blocking their synaptic transmission directly impairs spatial working memory [130,131]. Furthermore, PV+ GABAergic neurons in distal network structures are required for proper hippocampal and cortical oscillations. Disruptions in these oscillations can result in severe spatial memory impairment, exemplifying the importance of oscillatory network activity in shaping pyramidal cell output during cognitive processing. Consequently, any dysfunction in interneurons, as seen in DS, can significantly affect cognitive processes [132].

Interneurons can have a developmental role in patients with SCN1A mutations. In mice, for example, SCN1A expression coincides with the neurophysiological maturation of fast-spiking interneurons and the evolution of spatial cognition. PV+ fast-spiking interneurons undergo a drop in action potential breadth and an increase in firing frequency during the second and third postnatal weeks, changing them from slow to fast signaling cells [133]. The discovery that interneurons in SCN1A-mutant mice do not develop the narrow action potential width characteristic of mature cells suggests that SCN1A expression is important in this metamorphosis [122].

## 5. Novel Pharmacological Approaches for DS Management: What Is the Role of Precision Medicine?

### 5.1. Standard DS Management

Seizures are known to lead to ID, and SE may cause numerous hospitalizations and death. Since seizures are frequently pharmacoresistant in patients with DS [134], control of convulsive seizures should be prioritized over nonconvulsive seizures, given their greater impact on quality of life and higher association with SUDEP [135]. Several AEDs and surgical options are currently available for DS. Complete seizure control is typically unachievable in patients with DS, and the management goal should focus on reducing the risk of extended seizures and SE [54]. Measures to reduce seizure triggers are also recommended, including allowing the child to nap if tired, avoidance of high ambient temperature, prophylactic antipyretics for vaccination and illness, prophylactic benzodiazepines or antibiotics for febrile illness, and avoiding overexertion or flashing lights [42].

Most patients with DS require at least two AEDs to achieve seizure control. Both clobazam and VPA are widely recommended first-line options for patients with DS [54]. VPA, a branched-chain fatty acid, can be administered intravenously, orally, or rectally and is generally well-tolerated. VPA’s mechanisms of action include affecting the GABAergic system, inhibiting α-ketoglutarate dehydrogenase, GABA transaminase, and succinate semialdehyde dehydrogenase, and enhancing glutamate decarboxylase to increase GABA levels in plasma and several brain regions. Eventually, VPA can stimulate GABA receptors to interfere with sodium channels, modulate potassium and calcium conductance, affect serotoninergic and dopaminergic transmission, and alter cerebral metabolism [136]. All make VPA a very effective AED against a range of generalized and focal seizure types, including tonic–clonic, myoclonic, tonic, partial, and absence seizures [137]. The starting VPA dose is recommended between 10 and 15 mg/kg/day, divided into two or three doses, followed by gradual increases to target doses in the range of 25 to 60 mg/kg/day based on patients’ tolerability and clinical responses. Despite several advantages, VPA is uncommonly used as monotherapy in patients with DS. Three studies reported responder rates of 23–52% [138,139,140]. However, the literature and personal experience suggest that polypharmacy, instead of VPA alone, may be sufficient to control seizures in patients with DS. Notably, VPA may increase serum concentrations of lamotrigine, phenobarbital, and the metabolite 10,11-epoxide of carbamazepine, ethosuximide, and rufinamide because VPA can inhibit cytochrome P450s, uridine glucuronyl transferases, and epoxide hydrolase. Common side effects of VPA include abdominal cramps, diarrhea, impaired coagulation, nausea, neutropenia, vomiting, and weight gain. VPA may be associated with hepatotoxicity, pancreatitis, teratogenicity, and endocrine disturbance, such as increased total testosterone levels, menstrual abnormalities, obesity, polycystic ovary syndrome, and teratogenicity [136].

Clobazam (7-chloro-1-methyl-5-phenyl-1,5-benzodiazepine), first synthesized in 1966, has been marketed as an anxiolytic agent since 1975 and an AED since 1984 [141,142]. Clobazam is a benzodiazepine and has good oral tolerability, safety, and broad spectrum anti-seizure effects and weaker sedative and amnesic effects compared to other benzodiazepines [143]. Due to its excellent effects and safety, clobazam is now approved as an adjunctive treatment for epilepsy in more than 100 countries [144]. The anti-seizure activity of clobazam may be mediated through its GABA-A receptor agonist activity and its active metabolite, N-desmethyl-clobazam (norclobazam) [143].

The double-blind, placebo-controlled, efficacy and safety CONTAIN (ClObazam in patieNTs with LennoxGAstaut SyNdrome) trial showed that physicians’ and caregivers’ assessments indicated that clobazam significantly improved weekly drop seizure rates in Lennox Gastaut Syndrome (LGS) [145]. Another study involving patients with LGS receiving adjunctive clobazam showed sustained seizure-free and significant seizure improvements at stable dosages through three years of therapy [146]. These findings support the excellent anti-seizure effects of clobazam. Since AEDs can only partially control DS-related seizures [134], clinical trials using valproate, stiripentol, and clobazam only achieved a 71–89% responder rate in reducing seizure frequency in patients with DS [138,147]. Consistent with these findings, treating DS with a triple therapy of stiripentol, valproate, and clobazam resulted in about a twofold increase in clobazam, a 5–7-fold increase in norclobazam levels, and no change in plasma valproate levels [148]. Therefore, valproate, stiripentol, and clobazam are regarded as the “gold standard treatment”.

The initial clobazam dose is 0.2 mg/kg/day, divided into twice-daily doses, followed by gradual increases to reach a target dose of 0.3–1.0 mg/kg/day, up to a maximum of 2 mg/kg/ day. It is recommended that clobazam dose escalation should not exceed weekly since clobazam and norclobazam for concentrations in patients’ serum may reach a steady state after 5 and 9 days, respectively [149,150]. Adverse effects include upper respiratory infections, constipation, drooling, lethargy, somnolence, pyrexia, and respiratory depression, especially at high doses or with concomitant opioids [145,146].

Previous reports showed that clobazam and VPA led to significant seizure reduction in patients with DS when combined with stiripentol [139,151]. Stiripentol is an AED specifically developed to treat DS. It was granted orphan drug designation in the European Union in 2001 and the US in 2008. Stiripentol acts as a positive allosteric modulator of the γ-aminobutyric acid type A (GABAA) receptor, enhancing GABA’s inhibitory effect on neuronal activity [152]. The efficacy of stiripentol in DS was demonstrated in a pivotal clinical trial, which showed that 71% of the patients in the stiripentol group had a ≥50% reduction in seizure frequency during the two-month treatment period compared to only 5% in the placebo group [151]. Stiripentol or topiramate (started at 1–2 mg/kg/day, followed by gradual increases to reach a target dose of 5–10 mg/kg/day) should be combined with the clobazam and VPA regimen, especially when the first-line regimen fails to achieve adequate control [153]. In cases where the first- and second-line treatments for epilepsy are inadequate [42], the following medications may be considered: (1) Clonazepam: Initiate at 0.05 mg/kg, with a subsequent dose of 0.1 mg/kg every 6 h [154]. (2) Levetiracetam: Start with 10–20 mg/kg/day, increasing up to a maximum of 60–80 mg/kg/day. (3) Zonisamide: Administer 6–12 mg/kg/day [154]. (4) Ethosuximide: For children between ages 3 and 6, start with 125 mg twice daily. Children older than 6 years and adults should start with 250 mg twice daily, with a maximum dose of 1500 mg/day [155]. (5) Phenobarbital: Begin with 10–20 mg/kg/day. The maintenance dose should be 1–4 mg/kg for adults and 3–4 mg/kg for children and neonates [156]. 

A high-fat, low-carbohydrate ketogenic diet for one year was reported to achieve a ≥75% reduction in seizure frequency and severity in nearly 80% of children with DS [157]. The initial stiripentol dose varies in different regions. The starting dose is 20 mg/kg/day, up to a maximum of 1000 mg/day, in Japan, 50 mg/kg/day, up to a maximum of 3000 mg/day, in the USA, and 20–50 mg/kg/day, divided into two to three daily doses, in Europe [158,159]. In contrast, an expert recommendation is to initiate treatment at a lower dose of 10–15 mg/kg/day and gradually increase this dose to a target of 50 mg/kg/day over a period of 2 to 4 weeks [153]. 

While histopathological results may help uncover the genetic pathophysiology of DS-related epilepsy, studies on histopathological outcomes in DS are scarce [160]. Surgical options for these patients are limited and generally considered a last resort when pharmacological and dietary interventions have proven insufficient. A case series, while surgical interventions for other forms of epilepsy, such as resective surgery or the implantation of a vagus nerve stimulator, are effective, their applicability to DS is not as well-established due to their genetic and generalized nature [7]. Vagus nerve stimulation (VNS) has been used as a treatment option for patients with DRE, including a small subset of patients with DS. A few case reports and small-scale studies have reported varying degrees of success with VNS in patients with DS [161]. However, corpus callosotomy use has been documented in patients with DS, with mixed results [162]. There is limited evidence for the effectiveness of deep brain stimulation in DS [163].

Despite the plethora of available AEDs and trials of surgical options, the management of DS remains suboptimal since patients with DS may respond differently to various therapies. Therefore, personalized management is crucial to optimize treatment outcomes and ensure the best possible quality of life for individuals with DS.

### 5.2. Novel Pharmacological Approaches

#### 5.2.1. CBD

CBD is a non-psychoactive compound found in the cannabis plant that has gained significant attention for its potential therapeutic effects in various neurological disorders, including DS. The results of CBD treatment for DS have been promising, as demonstrated by two pivotal clinical trials. Based on their findings, the US Food and Drug Administration (FDA) approved CBD for DS in 2018 (Epidiolex^®^, GW Pharmaceuticals, Cambridge, UK) and the European Medicines Agency (EMA) approved CBD for DS in 2019 (Epidyolex^®^, GW Pharmaceuticals, Cambridge, UK).

In the first randomized, double-blind, placebo-controlled trial, 120 children and young adults with DS were enrolled and treated with either CBD or a placebo in addition to their existing AEDs. Its results showed a significant reduction in the median frequency of convulsive seizures per month in the CBD group compared to the placebo group. The median reduction in seizure frequency was 38.9% in the CBD group compared to 13.3% in the placebo group. Moreover, 5% of the patients in the CBD group became seizure-free during the trial, compared to none in the placebo group. Regarding safety, some adverse events were reported in both the CBD and placebo groups. The most common side effects in the CBD group included diarrhea, vomiting, fatigue, pyrexia, somnolence, and abnormal liver function tests. However, most of these side effects were considered mild to moderate in severity [164]. The other double-blind trial showed that CBD reduced the seizure frequency by 45.7–48.7% compared to 26.9% in the placebo group [165]. An open-label extension of the trial showed a median reduction in seizure frequency of 44% after a 48-week follow-up period [166]. The initial CBD dose is 5 mg/kg/day, which should be gradually increased to reach a target dose of 10 mg/kg/day within a minimum of 4 weeks, resulting in better tolerance and equivalent efficacy [167].

#### 5.2.2. Atypical Sodium Channel Blocker

Phenotypic variability and a wide range of *SCN1A* mutations in DS have led studies to investigate various treatment options. One such option is atypical sodium channel blockers, which target the neuronal voltage-gated sodium channels. These channels, particularly NaV1.1 encoded by *SCN1A*, are impaired by *SCN1A* mutations in patients with DS, leading to increased neuronal excitability and seizures. GS967 is one atypical sodium channel blocker that has been evaluated for DS. In a previous experimental model study, GS967 reduced seizure frequency and premature death in DS mice, which was attributed to the inhibition of sustained sodium currents and subsequent inhibition of synchronous repetitive action potential [168].

#### 5.2.3. Fenfluramine

Fenfluramine was initially used as an appetite suppressant in the 1960s but was withdrawn from the market due to cardiovascular valvular side effects [169]. However, due to its inhibition of the serotonin (5-hydroxytryptamine [5-HT]) transporter, the activation of multiple 5-HT receptors, the enhancement of 5-HT release into the synapse [170], the positive modulation of the sigma-1 receptor [171], and some promising results, low-dose fenfluramine was allowed for use in children with intractable epilepsy [169]. Furthermore, these results also support it as a potential treatment for DS [172,173]. The efficacy of low-dose fenfluramine in DS was demonstrated in a randomized, double-blind, placebo-controlled trial by Lagae et al. This trial enrolled 119 children and adolescents with DS experiencing drug-resistant seizures who were randomized to receive either fenfluramine or a placebo in addition to their existing AEDs. Its results showed a significant reduction in the monthly seizure frequency in the fenfluramine group (62.3%) compared to the placebo group (13.2%). Additionally, 54% of the patients in the fenfluramine group experienced a reduced seizure frequency of ≥50%, compared to only 5% in the placebo group [174]. This trial’s positive results led the FDA to approve fenfluramine in 2020 for treating seizures associated with DS in patients aged ≥2 years. Additionally, patients given fenfluramine had significantly longer seizure-free periods [174,175,176]. Fenfluramine use is reported to significantly reduce SE episodes [177,178,179,180]. Moreover, fenfluramine may reduce SUDEP mortality rates in patients with DS [181]. Fenfluramine is generally well-tolerated. Its common adverse effects include upper respiratory tract infection, decreased weight, diarrhea, fatigue, lethargy, and pyrexia. Fenfluramine is contraindicated in patients with aortic or mitral valvular heart disease and pulmonary arterial hypertension. The prescribing physicians must conduct periodic echocardiograms in patients taking fenfluramine to monitor their risk for heart disease and cardiac death [182,183]. If patients are co-administered with other serotonergic drugs, clinicians should be aware of the possibility of serotonin syndrome [182,183]. If patients also take VPA, clinicians should keep in mind that fenfluramine may exacerbate VPA-induced thrombocytopenia [184]. Fenfluramine is taken twice daily. However, an initial dose of 0.2 mg/kg/day is recommended, increasing to 0.4 mg/kg/day after seven days; 0.4 mg/kg/day is the recommended maintenance dose in patients taking stiripentol, which can increase plasma fenfluramine concentrations. If patients do not take stiripentol, a further increase to the suggested maintenance dose of 0.7 mg/kg/day can occur after another seven days [182].

#### 5.2.4. Other Agents

There are several agents currently being studied in patients with DS.

##### Soticlestat (TAK-935/OV935)

Soticlestat, a cholesterol 24-hydroxylase inhibitor [185], has been found to significantly reduce both spontaneous and hyperthermia-induced seizures and prevent premature death in a DS mouse model [186]. In a double-blind, placebo-controlled, multicenter, phase II study for DEEs, including DS (the ELEKTRA study; NCT03650452), soticlestat was administered to 51 pediatric patients with DS at a weight-adjusted equivalent of adult 300 mg BID for adults. This achieved a median placebo-adjusted seizure frequency reduction of 46% from baseline [187]. Common adverse events included lethargy and constipation. Additionally, a phase III, randomized, double-blind, placebo-controlled trial (the SKYLINE study; NCT04940624) is currently recruiting patients with DS aged 2–21 years to test whether soticlestat, used as an add-on therapy, can reduce seizure frequency.

##### Clemizole (EPX-100)

Clemizole, an old medication, was used to treat itches in the 1950s and 1960s. EPX-100, identified as having potential antiepileptic activity, was tested using a mutant zebrafish model of DS [188,189]. A placebo-controlled, double-blind, phase I study (NCT04069689), which involved 24 healthy volunteers, confirmed that EPX-100 was safe and well-tolerated in three groups of eight healthy adult individuals each [190]. Currently, a global, multicenter, randomized, double-blind, placebo-controlled phase II study (Argus Trial; NCT04462770) is recruiting individuals aged 2 years and older with DS to evaluate the safety and efficacy of EPX-100 as adjunctive therapy.

##### Lorcaserin (Belviq)

Lorcaserin, a serotonin receptor (5-HT2C) agonist, was initially approved by the FDA for treating obesity [191] and has been investigated for treating nicotine [192], opioid [193], and cannabis [194] use disorders. Based on findings that mice lacking 5-HT2C receptors were found to have a lower seizure threshold [195], lorcaserin has been found to reduce seizure activity in a zebrafish model of DS [196]. Two studies using lorcaserin in patients with DS showed a significant reduction in the frequency of seizures [195,197]. Lorcaserin is a compassionate use program for children with DS [198] and is currently in phase III development [199]. Common adverse events include headaches, dizziness, nausea, fatigue, and dry mouth. However, the FDA requested the withdrawal of this medication due to a potential risk of cancer that outweighs the benefits [200]. The safety of this medication deserves further evaluation.

##### Tiagabine

Tiagabine, which inhibits GABA reuptake into presynaptic neurons, has emerged as a potential treatment for DS. Tiagabine can increase the availability of GABA and enhance its inhibitory effect on neuronal activity. While tiagabine has been approved for the adjunctive treatment of partial seizures in adults and children aged ≥12 years, its role in DS has not been well-established. A synergism between tiagabine and clonazepam was evident in DS models, resulting in reduced side effects compared to those of the single drugs [201]. However, it is important to note that the evidence supporting tiagabine use in DS is limited and not based on rigorous clinical trials.

### 5.3. Gene Therapy

Monogenic disorders are a group of human diseases caused by mutations in single genes, which may reduce the expression of wild type proteins, leading to diseased phenotypes. Therefore, strategies to increase wild type proteins in subjects with such diseases may yield therapeutic benefits. Gene therapy applies genes expressing a therapeutic principle to treat and prevent specific diseases [202]. Monogenic diseases are ideal targets for gene therapy because the causative genes have been identified. Such therapies have been shown promising effects in treating monogenic diseases, such as spinal muscular atrophy (SMA), severe combined immunodeficiency (SCID), and hemophilia [203]. With more approved and under-approval gene therapy products on the market, and several gene therapy clinical trials in this field, the concept of fixing a broken gene by gene therapy is not inconceivable [204]. Gene therapies at least include gene interference, gene augmentation, and gene editing. Gene interference adjusts gene transcription or expression using ribonucleic acid (RNA) or small molecules to achieve inhibiting gain-of-function mutant expression through degrading mutant messenger RNA (mRNA) or correcting pre-mRNA splicing. Gene augmentation involves introducing an exogenous gene into cells to compensate for functional protein loss due to gene mutations. Gene editing seeks to repair mutated genes by modifying the genome using editing tools that are theoretically applicable to any mutations. Advances in those strategies in the modulation of target gene expression indicate a promising perspective for precise, causative treatment in DS.

#### 5.3.1. ASO Therapy

Antisense oligonucleotides (ASOs) are small fragments of synthetic genetic material that bind to RNA to correct the splicing of genes. ASO therapy has been proven to be successful in the treatment of SMA [205]. The *SCN1A* gene, implicated in DS, may contain an alternatively spliced poison exon with an in-frame stop codon [15,206]. ASOs complementary to poison exon block its inclusion through steric hindrance, thereby increasing the abundance of the full-length transcript [206]. In the context of DS, ASO therapy has been investigated as a potential treatment option due to its genetic nature (i.e., mutations in the *SCN1A* gene encoding the voltage-gated sodium channel NaV1.1). Preclinical studies have demonstrated that ASO therapy significantly ameliorates seizures and mortality in DS and has improved behavioral outcomes in treated mice, with brain NaV1.1 levels increased to become comparable to wild type mice [15].

STK-001, a compound produced by targeted augmentation of nuclear gene output (TANGO) using ASOs, can increase mRNA levels of the sodium channel Nav1.1, leading to optimal Nav 1.1 protein expression [206]. A single dose of STK-001 intraventricularly injected into the brain of a DS mouse model was found to increase the levels of Nav1.1 protein and reduce the incidence of SUDEP [15]. STK-001 has received orphan drug and rare pediatric disease status in the U.S. and orphan drug designation in Europe for treating patients with DS. A phase II trial (the MONARCH study; NCT04442295), using single or multiple ascending doses of STK-001, assessed safety, pharmacokinetics, and changes in seizure frequency. The results showed that ASO therapy was well-tolerable and reduced the frequency of convulsive seizures (with median reductions of 17–37%) [207]. While these findings are encouraging, further studies are needed to optimize the design and delivery of ASOs and assess the long-term safety and efficacy of this therapeutic approach in patients with DS.

#### 5.3.2. Gene Augmentation

To achieve optimal effects, a specific gene needs to be successfully delivered into a target cell. Several groups are designing numerous novel gene-delivering technologies for the treatment of treating DS, most of which are based on adeno-associated viral (AAV) vectors [208,209,210]. Among these, ETX-101 is a promising one-time, disease-modifying gene regulation therapy for DS. ETX-1 employs AAV-9 as a delivery method to transport functional copies of genes directly into target cells, compensating for the effects of genetic mutations. ETX-1 is also a cell-selective gene therapy because it contains an engineered transcription factor governed by a regulatory element specific to GABAergic cells. Theoretically, ETX-1 can increase the transcription and subsequent translation of the *SCN1A* gene specifically in inhibitory interneurons after a successful transfer. The therapy has been shown to be effective in decreasing the frequency and intensity of seizures during postnatal days 26–28 and extended survival up to 470 days after treatment in a mouse model of DS [16]. A phase 1/2, two-part, multicenter (the ENDEAVOR study; NCT05419492) trial is initiated to evaluate the safety and efficacy of ETX101 in participants with mutated *SCN1A* DS aged 6 to 36 months. Part 1 follows an open-label dose-escalation design, and Part 2 is a randomized, double-blind, sham delayed-treatment control dose-selection study. This trial is still ongoing.

#### 5.3.3. Gene Editing

Genome editing is a method that generates permanent modifications to DNA sequences at specific locations. Recent advances in gene editing tools include zinc-finger nucleases (ZFNs), transcription activator-like effector nucleases (TALENs), and clustered regularly interspaced short palindromic repeat-associated nucleases (CRISPR/CAS) [211]. ZFNs are based on zinc finger proteins, which are transcription factors, fused on the Foki endonuclease [212]. ZFNs can produce gene point mutations, deletions, insertions, inversions, duplications, and translocations in a complex genome and have been applied to cell and animal biotechnology with potential therapeutics [213]. TALENs are fusion proteins of a bacterial TALE (transcription activator-like effector) protein and the FokI endonuclease [214]. As the activity of each TALE domain is restricted to only one nucleotide and does not affect the binding specificity of neighboring TALEs, engineering TALENs is much easier than ZFNs, but the technology may have higher off-target binding [215]. The CRISPR/Cas9 system is composed of Cas9 endonuclease and two RNA molecules, crRNA and tracrRNA. crRNA and tracrRNA require an RNAseIII enzyme for processing and maturation of the crRNA molecule, which guides the Cas9 endonuclease to its DNA target [216]. Among these three systems, the CRISPR-Cas system is the easiest to design and implement and is highly predictable in experiment results. Therefore, it has become the most popular and favored tool for editing genomes. 

CRISPR-Cas9 technology can be applied to the treatment of haploinsufficient disorders because CRISPR is capable of increasing the expression of the wild type gene in the affected heterozygotes. Regarding the use of CRISPR-Cas9 in treating DS mouse models, two studies have been reported. Intraventricular injection of AAV carrying a transcriptional activator directed to the promoter of the *SCN1A* gene in the brain of the mice DS model resulted in elevated activity of inhibitory neurons in vivo and resistance to thermally induced seizures [208]. Conditional upregulation of the wild type *SCN1A* reduced seizure susceptibility and prolonged survival [217]. These studies, which apply the techniques to boost the expression of the wild type *SCN1A* gene specifically in inhibitory neurons, may make gene therapy a feasible treatment option for DS. A summary of standard and novel DS treatments is listed in Table 3.

## 6. Expert Perspectives and Future Directions

Precision medicine, an emerging approach that tailors medical treatment to an individual’s unique genetic, environmental, and lifestyle factors, has the potential to revolutionize DS management. A significant advance in understanding DS is the identification of the SCN1A gene as its primary cause in most cases. This discovery has motivated research into developing gene therapies that could potentially target and correct the underlying genetic defects. Targeted gene therapies can potentially provide a more effective and long-lasting solution for the management of patients with DS since these therapies address its root cause rather than only managing its symptoms. Several studies have shown that certain medications may be more effective in individuals with specific genetic variants, paving the way for tailored pharmacological interventions in DS. However, gene therapy for DS is still in its early stages, and further research is needed to determine its safety, efficacy, and feasibility in clinical settings.

Genetic testing plays a critical role in implementing precision medicine for DS management. Early and accurate DS diagnosis through genetic testing allows healthcare providers to promptly implement appropriate interventions, improving the overall prognosis and quality of life for affected individuals. Moreover, genetic testing can identify carriers of *SCN1A* mutations among family members, enabling informed reproductive choices and early intervention strategies for at-risk offspring. As our understanding of the genetic underpinnings of DS expands, the role of genetic testing in guiding precision medicine approaches will continue to grow.

## 7. Conclusions

Precision medicine has significant potential to improve the diagnosis and treatment of DS. Early genetic testing enables timely and accurate diagnosis. Advances in our understanding of DS’s underlying genetic mechanisms and neurobiology have enabled the development of targeted therapies, such as gene therapy or optogenetics, which might offer more effective and less invasive treatment options for patients with DS. Targeted therapies and gene therapies provide hope for more effective and personalized treatments. However, research into these therapeutic approaches is still in its early stages, and their clinical application remains to be seen.

## Figures and Tables

**Figure 1 ijms-25-00031-f001:**
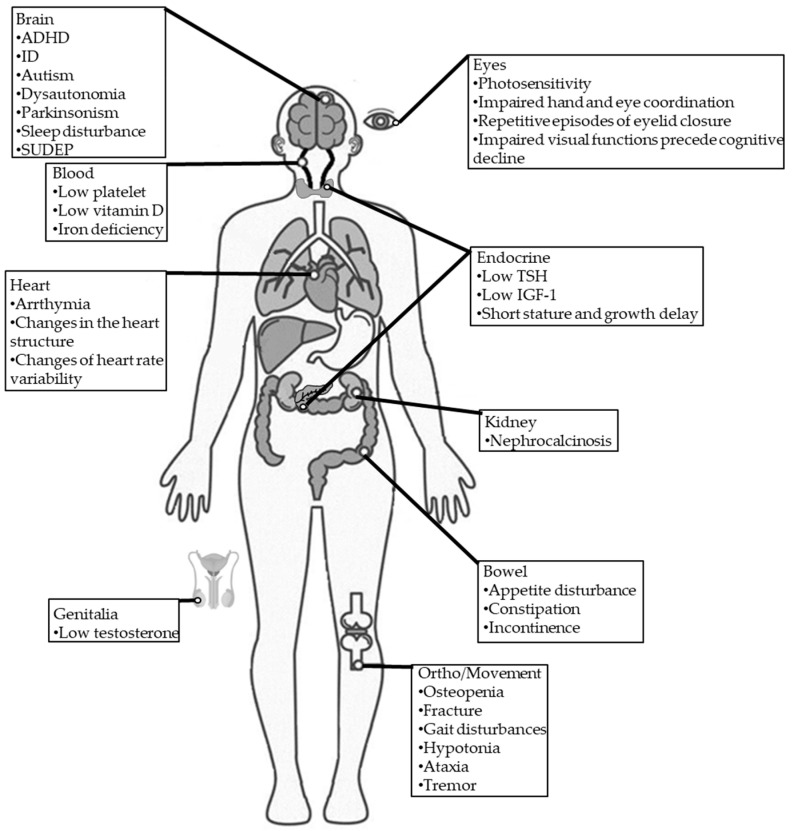
Comorbidities in individuals with DS. TSH: thyroid-stimulating hormone; IGF-1: insulin-like growth factor 1; SUDEP: sudden unexpected death in epilepsy.

**Table 1 ijms-25-00031-t001:** Various epilepsy syndromes fall within the DEE category.

DEE	Syndrome or Known Name	Phenotype MIM Number	Genes	Genetic Loci
DEE1	LISX2; XLAG; hydranencephaly with abnormal genitalia	300215	*ARX*	Xp21.3
	Proud syndrome	300004	*ARX*	Xp21.3
	PRTS; MRX36; MRXS1	309510	*ARX*	Xp21.3
	XLID29; MRX29; MRX32MRX33; MRX29; MRX38; MRX43; MRX52; MRX54; MRX76; MRX87	300419	*ARX*	Xp21.3
DEE2	EIEE2; ISSX2	300672	*CDKL5*	Xp22.13
DEE3	EIEE3	609302	*SLC25A22*	11p15
DEE4	EIEE4	612164	*STXBP1*	9q34
DEE5	EIEE5	613477	*SPTAN1*	9q34
DEE6A	EIEE6; SMEI;DS	607208	*SCN1A*	2q24.3
DEE6B	Phenotypes caused by heterozygous mutation in the SCN1A gene, not DS	619317	*SCN1A*	2q24.3
DEE7	EIEE7	613720	*KCNQ2*	20q13
DEE8	EIEE8; hyperkplexia and epilepsy	300607	*ARHGEF9*	Xq11.1
DEE9	EIEE9; EFMR; Juberg–Hellman syndrome	300088	*PCDH19*	Xq22.1
DEE10	EIEE10; MCSZ	613402	*PNKP*	19q13
DEE11	EIEE11	613721	*SCN2A*	2q24
DEE12	EIEE12	613722	*PLCB1*	20p12.3
DEE13	EIEE13	614558	*SCN8A*	12q13
DEE14	EIEE14	614959	*KCNT1*	9q34
DEE15	EIEE15	615006	*ST3GAL3*	1p34
DEE16	EIEE16	615338	*TBC1D24*	16p13
DEE17	EIEE17	615473	*GNAO1*	16q13
DEE18	EIEE18	615476	*SZT2*	1p34
DEE19	EIEE19	615744	*GABRA1*	5q34
DEE20	EIEE20; GPIBD4 MCAHS2	300868	*PIGA*	Xp22
DEE21	EIEE21	615833	*NECAP1*	12p13
DEE22	EIEE22; CDGIIm	300896	*SLC35A2*	Xp11
DEE23	EIEE23	615859	*DOCK7*	1p31
DEE24	EIEE24	615871	*HCN1*	5p12
DEE25	EIEE25	615905	*SLC13A5*	17p13
DEE26	EIEE26	616056	*KCNB1*	20q13
DEE27	EIEE27	616139	*GRIN2B*	12p12
DEE28	EIEE28	616211	*WWOX*	16q23
DEE29	EIEE29	616339	*AARS*	16q22
DEE30	EIEE30	616341	*SIK1*	21q22
DEE31A	EIEE31	616346	*DNM1*	9q34
DEE31B	-	620352	*DNM1*	9q34
DEE32	EIEE32	616366	*KCNA2*	1p13
DEE33	EIEE33	616409	*EEF1A2*	20q13
DEE34	EIEE34	616645	*SLC12A5*	20q12
DEE35	EIEE35	616647	*ITPA*	20p13
DEE36	EIEE36; CDG1s	300884	*ALG13*	Xq23
DEE37	EIEE37	616981	*FRRS1L*	9q31
DEE38	EIEE38; GPIBD23	617020	*ARV1*	1q42
DEE39	EIEE39; AGC1 DEFICIENCY	612949	*SLC25A12*	2q31
DEE40	EIEE40	617065	*GUF1*	4p12
DEE41	EIEE41	617105	*SLC1A2*	11p13
DEE42	EIEE42	617106	*CACNA1A*	19p13
DEE43	EIEE43	617113	*GABRB3*	15q11
DEE44	EIEE44	617132	*UBA5*	3q22
DEE45	EIEE45	617153	*GABRB1*	4p13
DEE46	EIEE46	617162	*GRIN2D*	19q13
DEE47	EIEE47	617166	*FGF12*	3q28
DEE48	EIEE48	617276	*AP3B2*	15q25
DEE49	EIEE49	617281	*DENND5A*	11p15
DEE50	EIEE50; CDG1Z	616457	*CAD*	2p23
DEE51	EIEE51	617339	*MDH2*	7q11
DEE52	EIEE52	617350	*SCN1B*	19q13
DEE53	EIEE53	617389	*SYNJ1*	21q22
DEE54	EIEE54	617391	*HNRNPU*	1q44
DEE55	EIEE55; GPIBD14	617599	*PIGP*	21q22
DEE56	EIEE56	617665	*YWHAG*	7q11
DEE57	EIEE57	617771	*KCNT2*	1q31
DEE58	EIEE58	617830	*NTRK2*	9q21
DEE59	EIEE59	617904	*GABBR2*	9q22
DEE60	EIEE60	617929	*CNPY3*	6p21
DEE61	EIEE61	617933	*ADAM22*	7q21
DEE62	EIEE62	617938	*SCN3A*	2q24
DEE63	EIEE63	617976	*CPLX1*	4p16
DEE64	EIEE64	618004	*RHOBTB2*	8p21
DEE65	EIEE65	618008	*CYFIP2*	5q33
DEE66	EIEE66	618067	*PACS2*	14q32
DEE67	EIEE67	618141	*CUX2*	12q23
DEE68	EIEE68	618201	*TRAK1*	3p25
DEE69	EIEE69	618285	*CACNA1E*	1q25
DEE70	EIEE70	618298	*PHACTR1*	6p24
DEE71	EIEE71; glutaminase deficiency with neonatal epileptic encephalopathy	618328	*GLS*	2q32
DEE72	EIEE72	618374	*NEUROD2*	17q12
DEE73	EIEE73	618379	*RNF13*	3q25
DEE74	EIEE74	618396	*GABRG2*	5q34
DEE75	EIEE75	618437	*PARS2*	1p32
DEE76	EIEE76; DECAM	618468	*ACTL6B*	7q22
DEE77	EIEE77; GPIBD19	618548	*PIGQ*	16p13
DEE78	EIEE78	618557	*GABRA2*	4p13
DEE79	EIEE79	618559	*GABRA5*	15q11
DEE80	EIEE80; GPIBD20	618580	*PIGB*	15q21
DEE81	EIEE81	618663	*DMXL2*	15q21
DEE82	EIEE82;GOT2 deficiency	618721	*GOT2*	16q21
DEE83	EIEE83; Barakat–Perenthaler syndrome	618744	*UGP2*	2p14
DEE84	EIEE84; Jamuar syndrome	618792	*UGDH*	4p14
DEE85	EIEE85	301044	*SMC1A*	Xp11
DEE86	EIEE86	618910	*DALRD3*	3p21
DEE87	EIEE87	618916	*CDK19*	6q21
DEE88	EIEE88	618959	*MDH1*	2p15
DEE89	-	619124	*GAD1*	2q31
DEE90	-	301058	*FGF13*	Xq26
DEE91	IECEE1	617711	*PPP3CA*	4q24
DEE92	IECEE2	617829	*GABRB2*	5q34
DEE93	-	618012	*ATP6V1A*	3q13
DEE94	EEOC	615369	*CHD2*	15q26
DEE95	GPIBD18	618143	*PIGS*	17q11
DEE96	-	619340	*NSF*	17q21
DEE97	-	619561	*iCELF2*	10p14
DEE98	-	619605	*ATP1A2*	1q23
DEE99		619606	*ATP1A3*	19q13
DEE100	-	619777	*FBXO28*	1q42
DEE101	-	619814	*GRIN1*	9q34
DEE102	-	619881	*SLC38A3*	3p21
DEE103	-	619913	*KCNC2*	12q21
DEE104	-	619970	*ATP6V0A1*	17q21
DEE105	-	619983	*HID1*	17q25
DEE106	-	620028	*UFSP2*	4q35
DEE107	-	620033	*NAPB*	20p11
DEE108	-	620115	*MAST3*	19p13
DEE109	-	620145	*FZR1*	19p13
DEE110	-	620149	*CACNA2D1*	7q21
DEE111	-	62054	*DEPDC5*	22q12.2–q12.3
DEE112	-	620537	*KCNH5*	14q23.2

ACC: agenesis of the corpus callosum; AGC1 deficiency: aspartate glutamate carrier 1 deficiency; CDG: congenital disorder of glycosylation; DECAM: developmental delay, epileptic encephalopathy, cerebral atrophy, and abnormal myelination; DS: Dravet syndrome; EEOC: epileptic encephalopathy, childhood onset; EFMR: epilepsy and mental retardation restricted to females; GOT2 deficiency: mitochondrial glutamate oxaloacetate transaminase deficiency; GPIBD: glycophosphatidylinositol biosynthesis defect; IECEE: epileptic encephalopathy, infantile or early childhood; ISSX2: infantile spasm syndrome, X-linked 2; LISX2: lissencephaly, X-linked, 2; MCSZ: microcephaly, seizures, and developmental delay; MCAHS: multiple congenital anomalies–hypotonia–seizures syndrome; MRX: mental retardation, X-linked; MRXS1: mental retardation, X-linked, syndromic 1; PRTS: Partington syndrome; SMEI: severe myoclonic epilepsy of infancy; XLAG: X-linked lissencephaly with ambiguous genitalia; XLID29: intellectual developmental disorder, X-linked 29; -: no available data. (Adopted and modified from https://omim.org/entry/308350; accessed on 22 November 2023).

**Table 2 ijms-25-00031-t002:** DS features in young children versus older previously undiagnosed children and adults.

Features	Children	Older Children and Adults
Neurodevelopment	Mostly normal at baseline Delays in gross motor, fine motor, language, and psychomotor ID ADHD	ID ADHD Crouch gait Some may show improvement in certain neurodevelopmental aspects, while others may continue to face challenges in behavioral, communication, and developmental problems
Triggering factors	Fever Infection Sun exposure/high ambient temperature Hot-water bath Intense physical exercise Overexcitement The use of sodium channel agents	Hyperthermia becomes less frequent The use of sodium channel agents
Seizure pattern and frequency	Seizure-free before onset First seizure appears before 7 mo. Frequent, prolonged hemiclonic or generalized seizures Multiple seizure types, including febrile or afebrile generalized/unilateral clonic/tonic seizures, myoclonic, and atypical absence. Focal seizures progress to focal motor or bilateral convulsive seizures NCSE is common More frequent and severe	Seizure patterns may evolve and may experience fewer fever-related triggers for seizures Generalized tonic–clonic seizure is the most common Myoclonic, focal, atypical absence, tonic seizures, SE, and NCSE becomes less frequent Gradually reduces in frequency and intensity but varies among individuals
EEG	Typically normal ≤ 1 y/o Generalized spike-and-wave patterns, polyspike waves, and multifocal spikes are common Some slow discontinuous symmetrical or asymmetrical patterns Photoparoxysmal response is common	Diffuse background slowing Focal epileptiform discharges over frontal or temporal regions Photosensitivity and pattern sensitivity typically disappear
Neuroimaging	Mostly normal Few show cortical malformations, HS, or a loss of gray-white volume	Normal Total intracranial volume decreased with age

**Table 3 ijms-25-00031-t003:** Standard and novel DS management.

Pharmacological Agents
First line: VPA
Second line: VPA + clobazam + stiripentol
Topiramate
Clonazepam
Levetiracetam
Zonisamide
Ethosuximide
Phenobarbital
Novel Pharmacological Approaches
CBD
Atypical sodium channel blocker
GS967
Fenfluramine
Other agents
Soticlestat
Clemizole
Lorcaserin
Tiagabine
Gene therapy
ASO
STK-001
Gene augmentation
ETX-101
Gene editing
CRISPR/Cas9
Surgery
Vagus nerve stimulation
Deep brain stimulation
Corpus callosotomy

ASO: antisense oligonucleotides; CRISP/Cas9: clustered regularly interspaced short palindromic repeat-associated nucleases; VPA: valproic acid.

## Data Availability

Not applicable because this review did not report any new data.

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
