# Peer review of "Clinical and Genetic Features of Dravet Syndrome: A Prime Example of the Role of Precision Medicine in Genetic Epilepsy"

_ijms, 2023, doi:10.3390/ijms25010031_

Round 1

Reviewer 1 Report

Comments and Suggestions for Authors

The manuscript “Clinical and Genetic Features of Dravet Syndrome: A Prime Example of the Role of Precision Medicine in Genetic Epilepsy” by Fan is a review article which addresses the current understanding of clinical Dravet syndrome (DS) features, genetic involvement in DS development, and outcomes of novel DS therapies. In particular, the authors focus on the roles of the sodium voltage-gated channel alpha subunit 1 (SCN1A) gene, which encodes a critical voltage-gated sodium channel subunit involved in neuronal excitability. In general, this review is critical in this field and contains essential contents. However, I have several comments before this manuscript is accepted for publication.

1. Figure 2 shows the typical structure of a voltage-gated Nav channel and does not provide a new information for the readers. At least, the authors should display SCN1A gene mutations associated with Dravet syndrome.

2. The section of pharmacological approaches is very interesting. Please summarize these information in the Table!

Author Response

Reviewer 1

Q: The manuscript “Clinical and Genetic Features of Dravet Syndrome: A Prime Example of the Role of Precision Medicine in Genetic Epilepsy” by Fan is a review article which addresses the current understanding of clinical Dravet syndrome (DS) features, genetic involvement in DS development, and outcomes of novel DS therapies. In particular, the authors focus on the roles of the sodium voltage-gated channel alpha subunit 1 (SCN1A) gene, which encodes a critical voltage-gated sodium channel subunit involved in neuronal excitability. In general, this review is critical in this field and contains essential contents. However, I have several comments before this manuscript is accepted for publication.

R: Thank you for your review.

Q1. Figure 2 shows the typical structure of a voltage-gated Nav channel and does not provide a new information for the readers. At least, the authors should display SCN1A gene mutations associated with Dravet syndrome.

 R: Thank you. The suggestion has been included in this revised manuscript.

Q2. The section of pharmacological approaches is very interesting. Please summarize these information in the Table!

R: The requested addition is now included in the updated manuscript.

Reviewer 2 Report

Comments and Suggestions for Authors

This is a comprehensive and interesting review on clinical and genetic aspects of Dravet syndrome. Current views on clinical features, pathomechanism and pharmacotherapy of this severe infantile-onset epileptic encephalopathy were discussed in detail. Furthermore, the authors provided some convincing  arguments that precision medicine has potential for improving the diagnosis and treatment of Dravet syndrome. In my opinion, this review is worth of publication after minor revision.

Specific remarks:

1.  The paper contains a great deal of relevant data, but it is not well ballanced. For example, the important  subchapter 5.3.2. AAV-9 Based Gene Therapy is too short. Therefore it should be supplemented with additional data, e.g. Yamagata T, Raveau M, Kobayashi K, Miyamoto H, Tatsukawa T, Ogiwara I, Itohara S, Hensch TK, Yamakawa K. CRISPR/dCas9-based Scn1a gene activation in inhibitory neurons ameliorates epileptic and behavioral phenotypes of Dravet syndrome model mice. Neurobiol Dis. 2020 Jul;141:104954.

2.     In its present form, the writing of this manuscript is not friendly to the readers. English editing and careful correction are required to the manuscript. There are significant differences in styles of the manuscript chapters, probably due to the fact, that several authors contributed to the article. The article should be stylistically uniform.

3.     Page 2: “Their outcomes are very poor even though seizures cease, and sometimes early death.” This sentence seems unfinished.

Author Response

Reviewer 2

This is a comprehensive and interesting review on clinical and genetic aspects of Dravet syndrome. Current views on clinical features, pathomechanism and pharmacotherapy of this severe infantile-onset epileptic encephalopathy were discussed in detail. Furthermore, the authors provided some convincing  arguments that precision medicine has potential for improving the diagnosis and treatment of Dravet syndrome. In my opinion, this review is worth of publication after minor revision.

Specific remarks:

Q:1.  The paper contains a great deal of relevant data, but it is not well ballanced. For example, the important  subchapter 5.3.2. AAV-9 Based Gene Therapy is too short. Therefore it should be supplemented with additional data, e.g. Yamagata T, Raveau M, Kobayashi K, Miyamoto H, Tatsukawa T, Ogiwara I, Itohara S, Hensch TK, Yamakawa K. CRISPR/dCas9-based Scn1a gene activation in inhibitory neurons ameliorates epileptic and behavioral phenotypes of Dravet syndrome model mice. Neurobiol Dis. 2020 Jul;141:104954.

R: Thank you. The requested supplemental information is now included.

  1. In its present form, the writing of this manuscript is not friendly to the readers.English editing and careful correction are required to the manuscript. There are significant differences in styles of the manuscript chapters, probably due to the fact, that several authors contributed to the article. The article should be stylistically uniform.

R: Thank you for your comment. We have made revisions as suggested.  

  1. Page 2: “Their outcomes are very poor even though seizures cease, and sometimes early death.” This sentence seems unfinished.

R: We revised the sentence to the following: In spite of the cessation of seizures, their outcomes may remain very poor, and early death has been reported in some cases.